# DOMINO: A DUAL-SYSTEM FOR MULTI-STEP VISUAL LANGUAGE REASONING

## ABSTRACT

Visual language reasoning requires a system to extract text or numbers from information-dense images like charts or plots and perform logical or arithmetic reasoning to arrive at an answer. To tackle this task, existing work relies on either (1) an end-to-end vision-language model trained on a large amount of data, or (2) a two-stage pipeline where a captioning model converts the image into text that is further read by another large language model to deduce the answer. However, the former approach forces the model to answer a complex question with one single step, and the latter approach is prone to inaccurate or distracting information in the converted text that can confuse the language model. In this work, we propose a dual-system for multi-step multimodal reasoning, which consists of a "System-1" step for visual information extraction and a "System-2" step for deliberate reasoning. Given an input, System-2 breaks down the question into atomic sub-steps, each guiding System-1 to extract the information required for reasoning from the image. Experiments on chart and plot datasets show that our method with a pre-trained System-2 module performs competitively compared to prior work on in- and out-of-distribution data. By fine-tuning the System-2 module (LLaMA-2 70B) on only a small amount of data on multi-step reasoning, the accuracy of our method is further improved and surpasses the best fully-supervised end-to-end approach by 5.7% and a pipeline approach with FlanPaLM (540B) by 7.5% on a challenging dataset with human-authored questions.

## 1 INTRODUCTION

Visual language reasoning for tasks such as question answering over charts/plots is computationally challenging: it requires (1) multi-step reasoning to decompose the original complex question, (2) extracting numbers or text from the information-dense images, and (3) performing arithmetic or logical reasoning to derive the final answer. Recent work on visual language reasoning has investigated both end-to-end and pipeline approaches. In the end-to-end approach (Lee et al., 2023; Liu et al., 2023b), a visual transformer is trained on a large amount of labeled data to answer questions based on images with a single step of inference. In the pipeline approach (Liu et al., 2023a), an off-the-shelf captioning model first converts the chart/plot into a linearized table. A text-only large language model (LLM) is then prompted to conduct chain-of-thought reasoning over the linearized table. The first approach empowers a unified model to accommodate both vision and language modalities, but struggles with questions requiring complex reasoning (Hoque et al., 2022). The second approach leverages the multi-step reasoning capabilities of LLMs on the verbalized table information. However, this conversion is prone to loss or distortion of information needed for reasoning (e.g., missing information about colors used in the plot). It also burdens the system unnecessarily as the linearized table is generated regardless of the question and thus may contain irrelevant information.

Inspired by the dual process theories of reasoning from cognitive science (Evans, 2003), in this work we present a dual-system for multi-step visual language reasoning called DOMINO. In particular, we use the notions of System-1 and System-2 processing in human brain introduced by Kahneman (2011), where System-1 corresponds to intuitive and habitual processing and System-2 refers to deliberate and controlled reasoning (Goyal & Bengio, 2022). Similarly, DOMINO alternates between two key modules, System-1 and System-2, to perform the task. In our context, System-1, realized as a visual reader, is responsible for intuitively extracting visual information from the image. System-2 which is implemented as an LLM reasoner, is responsible for more deliberate inference

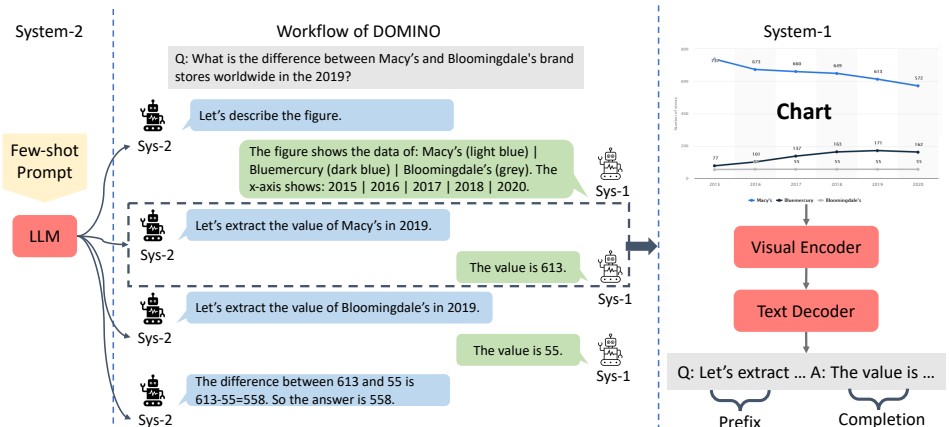

Figure 1: Overview of DOMINO, which alternates between System-2 (a prompted LLM) and System-1 (a visual encoder-text decoder) to answer complex questions over charts. The text in blue callouts are generated by System-2. The text in green callouts are generated by System-1 and appended to the generation sequence of System-2 directly. The chart and the question are from ChartQA (Masry et al., 2022).

by conducting multi-step reasoning for task decomposition, commonsense reasoning, and logical or mathematical operations for answer derivation. More specifically, given an image of a chart and a textual question (see Figure 1 for an illustration), System-2 decomposes the task into a sequence of steps (sub-tasks). For certain intermediate steps, System-1 is guided to obtain the visual information from the image. With the intermediate result extracted by System-1, System-2 either performs the next-step reasoning or derives the answer with all the available information. Throughout the process, DOMINO asks System-1 to obtain visual information when needed, instead of captioning the whole image at once, and thus allows more interactions between the two modalities.

We build System-1 based on a pre-trained visual language model Liu et al. (2023a) suited for chart understanding, which takes an image and a query from System-2 as input and returns the intermediate result. To further customize System-1 to the target data domain, we create a synthetic training set that contains different atomic operations over an image of a chart/plot (e.g., *extract the value of Macy's in 2019* from Figure 1) using templates. To implement System-2, we adopt an LLM in order to utilize its emergent reasoning capabilities. To adjust System-2 to conduct visual-language reasoning, we explore both few-shot prompting and fine-tuning with a handful of annotated examples.

We conduct experiments on several question answering tasks over charts/plots including ChartQA (Masry et al., 2022), PlotQA (Methani et al., 2020), DVQA (Kafle et al., 2018) and FigureQA (Kahou et al., 2018). The results show that without fine-tuning the LLM, DOMINO outperforms the pipeline approach using few-shot methods on both in- and out-of-distribution data. With only 100 training examples, DOMINO even outperforms the best fully-supervised method by 5.7% in accuracy on ChartQA that requires more deliberate reasoning. Further analysis shows that: (1) The intermediate results are essential to the success of our method. (2) System-2 benefits more from learning task decomposition when the questions are arbitrary and natural but more from learning answer deduction when the questions are restricted in type. (3) DOMINO is more robust in handling complex charts. (4) Fine-tuning the LLM on interacting with vision is more data-efficient than fine-tuning the LLM on table reasoning.

## 2 RELATED WORK

Visual language reasoning and question answering is an active area of research (Kafle et al., 2018; Kahou et al., 2018; Chaudhry et al., 2019; Methani et al., 2020; Masry et al., 2022). This is a special case of multimodal reasoning tasks that requires understanding an information-intense image and performing multi-step arithmetic or logical reasoning to derive an answer to complex questions. Recent work on visual language question answering focusing on charts/plots has investigated both supervised end-to-end and pipeline approaches.

**Supervised VQA.** Among supervised approaches, PReFIL (Kafle et al., 2020) allows for OCR integration and uses different recurrent and dense models to encode text and image inputs separately which are then fused and fed to a classifier for obtaining an answer. More recently, visual transformers are trained on a large amount of labeled data to answer questions based on images with different training objectives. PaLI (Chen et al., 2023b) and PaLI-X (Chen et al., 2023a) use OCR-aware pretraining objectives where the model predicts texts obtained from some OCR system. Similarly, ChartBERT (Akhtar et al., 2023) uses OCR text and positions to train a transformer encoder. Using OCR systems, however, adds computational cost and falls short on cases where the charts/plots do not have numbers and texts written explicitly (Liu et al., 2023b). Pix2Struct (Lee et al., 2023) and MATCHA (Liu et al., 2023b) are end-to-end models for visual language, where Pix2Struct provides generic checkpoints for different visual language tasks and MATCHA further fine-tunes Pix2Struct with new pretraining objects for chart derendering and mathematical reasoning. ChartT5 (Zhou et al., 2023b) learns to interpret table information from chart images via cross-modal pre-training on plot table pairs with masked header prediction and masked value prediction pre-training objectives. UniChart (Masry et al., 2023) also considers an encoder-decoder architecture and considers different pretraining objectives for low-level and high-level tasks. These approaches empower a unified model to accommodate both vision and language modalities, but struggle with questions requiring complex reasoning (Hoque et al., 2022). Unlike the supervised approches that force the model to answer the question with one single step, DOMINO leverages an LLM for multi-step reasoning.

**Pipelined VQA.** The pipeline approach, on the other hand, divides the task into two steps consisting of 1) information extraction or chart derendering and 2) question answering. There are different approaches to extract information from charts: some approaches combine OCR, object detection/segmentation techniques and/or heuristic rules for extracting information (Jung et al., 2017; Balaji et al., 2018; Luo et al., 2021; Akhtar et al., 2023), while others use a deep model to either extract different chart components (Cheng et al., 2023a) or convert the input chart to a textual table (Liu et al., 2023a). The resulting output is then reasoned over for question answering using table-to-text models (Andrejczuk et al., 2022), specialized models (Cheng et al., 2023a), large language models (Chen, 2023; Liu et al., 2023a), or code models combined with program executors (Chen et al., 2022; Cheng et al., 2023b). Different from these approaches, DOMINO interleaves and alternates between information extraction (System-1) and task decomposition and reasoning (System-2).

# 3 DOMINO

DOMINO is a dual-system for multi-step visual language reasoning. Unlike the end-to-end approach that uses one unified model to answer questions with one single step, DOMINO leverages an LLM to solve questions that require multi-step reasoning. Unlike the few-shot pipeline approach that converts a whole chart into a table, DOMINO only obtains information from the chart contextualized by one reasoning step at a time, thus allowing more interactions between the textual and visual modalities.

Figure 1 illustrates the workflow of DOMINO. Given an image of the chart $c$ and a textual question $q_0$, DOMINO alternates between both modules to deduce the answer $a$ step by step. Taking the original question $q_0$, and potentially the previous reasoning steps as input, System-2 either generates the next query $q_i$ ($i > 0$) to System-1 to obtain an intermediate result $a_i$ or answers the question by synthesizing all the intermediate results $\{q_i, a_i\}$. Guided by System-2, System-1 takes the chart $c$ and the query $q_i$ as input and returns the intermediate result $a_i$. We describe each module below.

## 3.1 SYSTEM-1

Our System-1 is responsible for the intuitive part of visual language reasoning, i.e., extracting information from the chart/plot. We implement System-1 as a vision encoder-text decoder Transformer model (Lee et al., 2023). Given a chart $c$ and a textual query $q_i$ from System-2, the visual encoder first represents the chart as a sequence of patch embeddings. Then the query $q_i$ is fed as the prefix of the text decoder to guide System-1 to generate the answer $a_i$ by decoding from $P(a_i|c, q_i)$.

**Atomic Operations** We define the following list of atomic operations that are needed to extract information from charts/plots in general. These operations facilitate the interaction between System-

2 and System-1. System-2 can choose to flexibly combine these atomic operations according to the reasoning structure, whereas System-1 can execute these operations individually.

- `Describe`: Since System-2 is a text-only LLM and does not access the chart directly, it is challenging for System-2 to generate valid queries that are grounded to the chart. For example, it may ask for a data point that does not exist in the chart. To avoid such hallucinating behavior, we define our first atomic operation `Describe`, which allows System-2 to get a high-level description of the chart. When receiving this query, System-1 responds with the key elements that are visualized in the chart (see the first green callout from System-1 in Figure 1 for an example). We let System-2 always use `Describe` as the first reasoning step since it is vital for System-2 to ask valid queries in the following steps.

- `Extract-Point`: This atomic operation is designed to allow System-2 to obtain the value of a specific data point, e.g., *extract the value of Macy's in 2019*, which is usually required for questions like "*What is the difference between Macy's and Bloomingdale's in 2019?*". When receiving this query, System-1 only needs to extract one single value from the chart, which is more efficient than the pipeline approach which would extract all values.

- `Extract-Group`: The last atomic operation is designed to allow System-2 to obtain the values of a certain group, e.g., *extract the value of Macy's*, which is required for questions like "*What is the maximum value of Macy's across all years?*". When receiving this query, System-1 returns all the values of *Macy's*.

**Training** For each type of atomic operation, we generate the query-answer pairs $\{q_i, a_i\}$ automatically based on available annotated data using templates (detailed in the experiment section). Table 6 in appendix § A.1.2 shows the examples of these query-answer pairs. We then train System-1 by applying the standard language modeling loss on the answer spans.

## 3.2 SYSTEM-2

Answering questions over charts/plots usually involves complex reasoning such as arithmetic and logical operations (taking the sum, finding the maximum, comparing values, etc.) (Masry et al., 2022). Due to their strong capability in step-by-step reasoning Wei et al. (2022), we adopt LLMs as System-2 for task decomposition.

**Workflow** Figure 1 illustrates the whole workflow of System-2. Given an originally complex question $q_0$, and optionally the previous reasoning steps $\{q_i, a_i\}$, System-2 can select one of the atomic operations to ask a further query $q_{i+1}$ from System-1 or deduce the answer $a$ as the final step. If a further query $q_{i+1}$ is generated, e.g., "*Let's extract the value of Bloomingdale's in 2019.*", we feed $q_{i+1}$ alone to System-1 for obtaining the intermediate result $a_{i+1}$, e.g., "*The value is 55*". Then we append $a_{i+1}$ back to the current generation sequence of System-2 which continues to generate the next step. If System-2 acquires all the required information, it would conduct chain-of-thought reasoning to synthesize all the information to deduce the final answer $a$, e.g., "*The difference between ... So the answer is 558.*".

**Learning to Decompose** We now describe how to adapt an LLM to compose the atomic operations defined above to collect all the information required for answering a complex question over a chart. We explore both *prompting-only* and *prompting+fine-tuning* as two means of adaptation.

- *Prompting-only*: We use few-shot prompting to adapt an LLM to conduct visual language reasoning by interacting with System-1. In the prompt, each example consists of a question $q$, the intermediate reasoning steps $\{q_i, a_i\}$, and finally a concluding sentence ending with the answer $a$ (see Figure 1 for the format of the prompt and Appendix § A.2 for the full prompt we used).

- *Prompting+fine-tuning*: Recent works show that with minor fine-tuning, the reasoning capability of an LLM can be greatly enhanced (Yu et al., 2023; Zhou et al., 2023a). We take inspiration from this observation and study how much we can improve the performance of DOMINO by fine-tuning System-2 with only a few ($<= 100$) training examples. Through fine-tuning, we aim to teach System-2 to both (1) decompose the task and (2) deduce the answer. During training, we only apply the language modeling loss on the text that is supposed to be generated by System-2 during the inference time — i.e., the query spans $q_i$ and

the final concluding sentence leading to $a$. This is to avoid teaching System-2 to hallucinate the parts that should be generated by System-1, i.e., the intermediate answers $a_i$. During inference, we still provide the few-shot prompt to System-2 as we find this leads to a better performance of DOMINO overall.

# 4 EXPERIMENTAL SETUP

## 4.1 DATASETS

For fine-tuning and evaluation we use the ChartQA (Masry et al., 2022) and PlotQA (Methani et al., 2020) datasets. ChartQA has two subsets. One is machine generated (marked with *augmented*) and the other is human written (marked with *human*) which requires more complex reasoning. PlotQA also has two sets: v1 (mostly focused on extractive questions) and v2 (requires more numerical reasoning), both of which are machine generated. Details of each dataset are reported in Appendix A.1. For fine-tuning System-1, we use samples from training sets of these datasets along with templates for each of the `Describe`, `Extract-Point`, and `Extract-Group` atomic operations to generate the training data. See Appendix A.1.2 for examples of templates and generated data. For fine-tuning System-2, we collect 100 high-quality question decomposition and reasoning examples. More specifically, we sample diverse charts/questions from the training sets of ChartQA and PlotQA and ask an annotator to decompose each complex question into atomic operations and deduce the answer. See Appendix A.1.3 for examples of the collected data.

Since parts of the training sets of ChartQA and PlotQA are used during the fine-tuning stage, we also evaluate DOMINO on two additional datasets that were not used during fine-tuning: DVQA (Kafle et al., 2018) and FigureQA (Kahou et al., 2018). Both of these datasets include chart images from synthetic tables that are randomly generated from limited vocabularies. FigureQA has yes/no answers whereas DVQA contains open ended questions where many refer to texts specific to the corresponding charts. While DVQA only includes bar charts, FigureQA additionally includes line graphs and pie charts. For all synthetic datasets (i.e., PlotQA, DVQA, and FigureQA), we randomly sample 10K examples and use this set for evaluation.

## 4.2 TRAINING DETAILS

For System-1, we use DePlot as the backbone visual language model and fine-tune it on the synthetic dataset we created for atomic operations. We generate a total of $774,019$ examples using templates with the ChartQA and PlotQA training sets ($17,014$ for `Describe`, $362,955$ for `Extract-Point`, and $273,657$ for `Extract-Group`. In Appendix A.1.2 we have provided some examples of the generated data. We set the batch size as $256$, the learning rate as $1e-5$ and the training steps as 10K. For System-2, we use the 70B variant of the recently published LLaMA-2 (Touvron et al., 2023) family of models. Since we only use a handful of expert-annotated training examples ($<= 100$), we use a very small batch size of $8$ and set the learning rate as $1e-6$. We train for a maximum optimization steps of $20$ and apply the language modeling loss on the text generated only by System-2 as discussed in §3.2.

## 4.3 BASELINE MODELS AND EVALUATION METRICS

To evaluate the ability of DOMINO for answering complex questions about charts/plots, we compare it with several fully-supervised end-to-end approaches as well as the pipeline approach that first converts the chart/plot to a table and then reasons over the table step-by-step. Similar to prior work (e.g., (Liu et al., 2023a)), we report "relaxed accuracy" which computes exact match for textual responses but allows a $5\%$ tolerance for numeric answers. We compare DOMINO against the following strong baselines:

**Fully-Supervised** We consider the following state-of-the-art supervised approaches which were discussed in §2: ChartT5 (Zhou et al., 2023b), Pix2Struct (Lee et al., 2023), MATCHA (Liu et al., 2023b), UniChart (Masry et al., 2023), and PaLI-X (Chen et al., 2023a).

**Few-shot DePlot** DePlot (Liu et al., 2023a) is a pipeline approach where a model is first trained to translate an image to a textual table, and then different LLMs are used to reason over the table via

Table 1: Main results of the compared methods on downstream tasks. Best numbers are in bold and second best numbers are underlined. We re-evaluate the DePlot model with GPT-3 on our sampled subsets of PlotQA (marked by *). The results for other baselines (if available) are from the papers cited in the table. The 1-Shot prompt used in DePlot consists of 1 table with 5 question-answer pairs, while the 5-Shot prompt we use consists of 5 tables with 1 question-answer pair each.

| Method | ChartQA | | | PlotQA | | |
|---|---|---|---|---|---|---|
| | Aug. | Human | Avg. | V1 | V2 | Avg. |
| **Fully-Supervised** | | | | | | |
| ChartT5 (Zhou et al., 2023b) | 74.4 | 31.8 | 53.2 | - | - | - |
| Pix2Struct (Lee et al., 2023) | 81.6 | 30.5 | 56.1 | 73.2 | 71.9 | 72.6 |
| MATCHA (Liu et al., 2023b) | 90.2 | 38.2 | 64.2 | **92.3** | **90.7** | **91.5** |
| UniChart (Masry et al., 2023) | 88.6 | 43.9 | 66.2 | - | - | - |
| PaLI-X (Chen et al., 2023a) | - | - | 70.9 | - | - | - |
| PaLI-X with OCR (Chen et al., 2023a) | - | - | 72.3 | - | - | - |
| **Few-Shot DePlot** | | | | | | |
| GPT3 (1-Shot) (Liu et al., 2023a) | 37.3 | 36.5 | 36.9 | *31.6 | *42.2 | *36.9 |
| FlanPaLM (540B) (1-Shot) (Liu et al., 2023a) | 76.7 | 57.8 | 67.3 | 51.3 | 44.9 | 48.1 |
| FlanPaLM (540B) (1-Shot, SC) (Liu et al., 2023a) | 78.8 | 62.2 | 70.5 | 57.8 | 50.1 | 53.9 |
| LLaMA-2 (70B) (1-Shot) | 86.5 | 53.5 | 70.0 | 32.5 | 43.4 | 37.9 |
| GPT4 (5-Shot) | 83.8 | 61.4 | 72.6 | - | - | - |
| LLaMA-2 (70B) (5-Shot) | 87.4 | 59.4 | 73.4 | 43.2 | 44.7 | 43.9 |
| **Other Pipeline Approaches** | | | | | | |
| ChartReader (Cheng et al., 2023a) | - | - | 52.6 | 78.1 | 59.3 | 68.7 |
| **DOMINO (our method)** | | | | | | |
| LLaMA-2 (70B) (5-Shot) | 88.6 | 59.3 | 74.0 | 53.1 | 59.0 | 56.1 |
| − without `Describe` | 77.4 | 45.6 | 61.5 | 40.5 | 62.7 | 51.6 |
| LlaMa-2 (70B) (5-Shot, SC) | 90.3 | 61.4 | 75.8 | 57.3 | 71.3 | 64.3 |
| Fine-tuned LLaMA-2 (70B) (5-shot) | 91.7 | 61.7 | 76.7 | 55.1 | 71.3 | 63.2 |
| Fine-tuned LLaMA-2 (70B) (5-shot, SC) | **91.8** | **64.1** | **78.0** | 58.9 | 80.7 | 69.8 |

few-shot learning with Chain-of-Thought (CoT) prompting (Wei et al., 2022). We compare against this model with the following LLMs: GPT3 (Brown et al., 2020), FlanPaLM (540B) (Chung et al., 2022), LLaMa-2 (70B) (Touvron et al., 2023), and GPT4 (OpenAI, 2023). Following Liu et al. (2023a), we adopt both (1) sampling and (2) self-consistency (SC) decoding (Wang et al., 2023), which samples a set of generations and chooses the majority-voted answer, and use a temperature of 0.4. The 1-Shot prompt used in DePlot consists of 1 table with 5 question-answer pairs. However, this may mislead the LLM to assume that the new question is from the same context since we do not have any tables in the prompt. To align with our method, we also experiment with a 5-Shot prompt consisting of 5 tables, each with 1 question-answer pair (see Appendix A.2).

## 5 MAIN RESULTS

Table 1 reports the results of comparing our method against fully-supervised and pipeline methods on ChartQA and PlotQA. We observe that: (1) Without fine-tuning, DOMINO already outperforms the best fully-supervised method (PaLI-X with OCR) on the ChartQA dataset (72.3% → 75.8%), where the questions are more diverse and complex. This demonstrates the effectiveness of DOMINO in handling such questions by leveraging the strong language understanding and task decomposition capabilities of the LLM. The fully-supervised methods do perform better than both DePlot and DOMINO on PlotQA. This is because PlotQA is a synthetic dataset with template-based and restricted types of questions. The fully-supervised methods can learn the bias in data encoded in the large training set (with over 100M examples) as pointed out by Liu et al. (2023a). (2) DOMINO also outperforms DePlot using either GPT3, LLaMA-2 (70B) or the much larger FlanPaLM (540B) model on both ChartQA and PlotQA, and DePlot with GPT4 on ChartQA[1]. This demonstrates the benefits of DOMINO which allows more interactions between the language and the vision components, and does not introduce redundant information as DePlot does when convert-

---

[1]We did not run GPT4 on the much larger PlotQA evaluation sets due to high cost and usage limits.

Table 2: Experimental results of the compared methods on the out-of-distribution datasets. Best numbers are in bold and the second best numbers are underlined. Our results are reported on 10K random sample of the corresponding evaluation sets. The results for other baselines are from their papers as cited in the table.

| Method | DVQA | FigureQA | |
|---|---|---|---|
| | Test-Novel (Reasoning) | Val1 | Val2 |
| **Seen at Training** | | | |
| **State-of-the-Art** | | | |
| PReFIL (no OCR) (Kafle et al., 2020) | 49.2 | - | - |
| PReFIL (with OCR) (Kafle et al., 2020) | **80.7** | - | - |
| ChartReader (Cheng et al., 2023a) | - | **95.5** | **95.8** |
| **Unseen at Training** | | | |
| **Few-Shot DePlot** | | | |
| LLaMA-2 (1-Shot) | 40.3 | 55.6 | 55.7 |
| LLaMA-2 (5-Shot) | 54.2 | 61.6 | 61.2 |
| **DOMINO (our method)** | | | |
| LLaMA-2 (5-Shot) | 55.2 | 63.2 | 62.7 |
| Fine-tuned LLaMA-2 (5-shot) | 55.4 | 64.7 | 64.4 |

ing a chart into a table[2]. (3) With minor fine-tuning using only a handful of 100 examples annotated with the reasoning process, we can further improve the performance of DOMINO on both ChartQA and PlotQA. We study data efficiency in § 6. Notably, with self-consistency decoding, DOMINO outperforms the best fully-supervised method by 5.7% in accuracy on ChartQA and we also observe a large performance boost ($71.3\% \rightarrow 80.7\%$) on PlotQA-V2 which contains more numerical reasoning questions.

In Table 2, we report the results of DePlot and DOMINO on out-of-distribution (OOD) datasets including DVQA and FigureQA, and compare them with fully-supervised methods. Here the OOD setting means that neither System-1 nor System-2 of DOMINO is fine-tuned on the experimented datasets. DOMINO does not outperform the supervised methods due to the synthetic nature of these datasets and the fact that both PReFIL and ChartReader were fine-tuned on the training partitions of DVQA and FigureQA, respectively. However, with regard to few-shot approaches, results show that DOMINO generalizes better than DePlot. Future work could enhance DOMINO with a more advanced vision module to improve generalization capabilities.

## 6 ANALYSIS & DISCUSSION

**Effectiveness of image description in addressing hallucination** In this ablation study, we investigate the effectiveness of the `Describe` operation in providing the initial context to System-2 so that System-2 asks valid queries afterwards. We prompt System-2 with examples where `Describe` is not used at all. The performance of the resulting DOMINO variant is shown in Table 1 (without `Describe`). We observe that discarding the `Describe` step generally leads to a considerable performance drop of DOMINO except on the PlotQA-V2 split. This demonstrates the effectiveness of the `Describe` step in providing the necessary context for System-2 to generate the right decomposition steps, especially when the questions are flexible in terms of wording and may not provide enough information for reasoning as the synthetic questions from PlotQA-V2 do.

**Skills learnt from fine-tuning System-2** We see significant improvement by fine-tuning System-2 in Table 1 and would like to investigate how the skills learnt from fine-tuning, i.e., task decomposition and answer deduction, contribute differently to the overall performance. We fine-tune System-2 by applying the language modeling loss only on (1) the intermediate queries $\{q_i\}$ or (2) the concluding sentence leading to the final answer $a$. The results are shown in Table 3, where we have opposite observations on ChartQA and PlotQA, which reveals that the supervision on the intermediate process is not always beneficial. On ChartQA, we see a larger performance drop from fine-tuning

---

[2]DePlot reports the best results on ChartQA (79.3%) using Codex (Chen et al., 2021) with Program-of-Thoughts (PoT) (Chen et al., 2022) and SC. However, we note that our method could also be adapted to work with code-LM which we have not investigated in this work.

Table 3: Ablation study on how the task decomposition and answer deduction skills learnt in fine-tuning contribute differently to the overall performance. Method indicates what the language modeling loss was applied to.

| Method | ChartQA | | | PlotQA | | |
|---|---|---|---|---|---|---|
| | Aug. | Human | Avg. | V1 | V2 | Avg. |
| **Fine-tuned LLaMA-2** | | | | | | |
| Answering Steps | 86.3 | 48.1 | 67.2 | 51.1 | 76.8 | 64.0 |
| Decomposition Steps | 87.6 | 56.6 | 72.1 | 48.5 | 76.0 | 62.3 |
| Answering and Decomposition Steps | 91.7 | 61.7 | 76.7 | 55.1 | 71.3 | 63.2 |

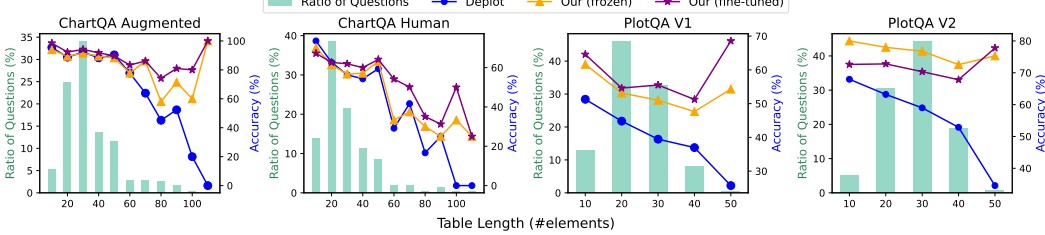

Figure 2: Performance grouped by the complexity of the underlying tables of the charts. The x-axes show the length of the underlying table of a chart. The left y-axes show the ratios of the questions in each length interval indicated by the green bars.

System-2 only on answer deduction. This demonstrates that LLMs struggle with task decomposition more than answer deduction when the questions are more natural. On PlotQA, however, we see a larger performance drop coming from fine-tuning System-2 only on decomposition steps for V1 and even performance gains from fine-tuning System-2 only on answer deduction or decomposition steps for V2. This is because the question types in PlotQA are rather restricted and in this case the LLM benefits more from just learning how to deduce the answer.

**Robustness in handling complex charts** We investigate whether the multi-interplay between language and vision allows DOMINO to perform robustly on more complex charts. Here, we use the length of the underlying table of a chart as a measurement of its complexity, and accordingly group the accuracy scores of DePlot and DOMINO (with a frozen or fine-tuned System-2) by the table length as shown in Figure 2. We observe that DOMINO (either frozen or fine-tuned) performs consistently better than DePlot on increasingly complex charts. This verifies the downside of converting charts to tables before reasoning as done in DePlot as it introduces redundant information and is error-prone, especially when the chart is very complex. DOMINO does not have this issue as we only require System-1 to obtain the necessary information required by one reasoning step.

**Data efficiency of reasoning-based fine-tuning** We study how DOMINO performs across different amount of training data. As comparison, we also fine-tune the LLM in DePlot on the same examples but annotated with chain-of-thought on tables. The results are shown in Figure 3. We observe that fine-tuned DOMINO generally outperforms fine-tuned DePlot across different numbers of training examples. One thing to note is that we do need sufficient examples to elicit the reasoning capabilities from the LLM so that it can outperform the frozen LLM (over 50 for PlotQA-V1 and over 20 for PlotQA-V2). Meanwhile, fine-tuning System-2 does not hurt DOMINO's OOD performance on DVQA, regardless of the number of training examples. In comparison, fine-tuning DePlot leads to worse OOD performance when more than 10 training examples are used.

**Inference efficiency** Although there may be multiple calls to the vision module in DOMINO, DOMINO is more efficient compared to the few-shot DePlot model due to two reasons: (1) The vision module in DOMINO only needs to generate the required information based on the image while the vision module in DePlot needs to generate the whole table, which can be arbitrarily long. (2) As a result of (1), DOMINO does not need to take as input the whole table sequence which would consume a large part of the context window in the LM of DePlot.

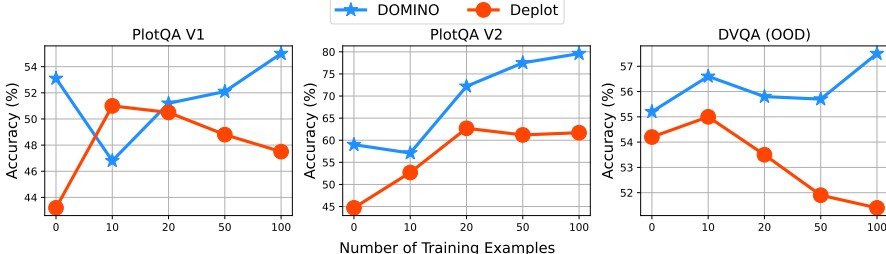

Figure 3: Ablation study on how number of fine-tuning examples affects performance of the LLM. The LLM in both methods are not fine-tuned when the number of training examples equals to 0.

Table 4: Case studies on ChartQA-human. The underlined text represents the errors made by the models. See Appendix A.3 for associated charts.

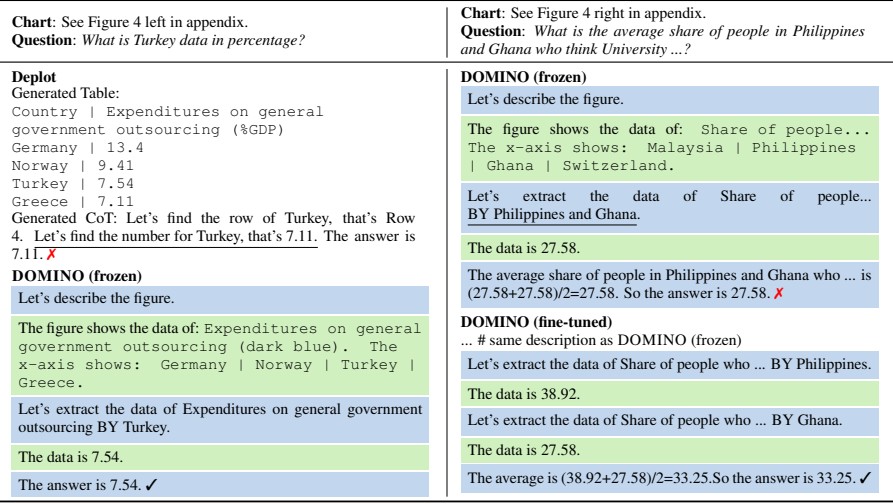

**Error Analysis** Since PlotQA is a synthetic dataset, we have information about the template types that were used to generate questions. Appendix Table 7 shows the breakdown of errors per template type for the PlotQA-V2 dataset. Across all template types we see that DOMINO yields improvements, but the most significant reductions in errors are for questions that require reasoning (i.e., arithmetic, compound, comparison, min-max) where we see reductions of $45\%$ to $68\%$ in errors when comparing the fine-tuned model against the DePlot model.

To illustrate the difference between different models, Table 4 shows examples from the ChartQA-human set. Table 4 (left) shows an example where DePlot correctly predicts the underlying table of the chart yet fails to extract the right value from the table due to the redundant information. By contrast, DOMINO only extracts the necessary information by generating a specific query to System-1 and thus answers correctly. Table 4 (right) shows an example where System-2 of DOMINO fails to leverage the information from the previous reasoning step (that `Philippines` and `Ghana` are two data groups in the chart) and thus generates an invalid query to System-1. Through fine-tuning, System-2 learns to properly decompose the question and generates the right queries to obtain the intermediate results.

## 7    CONCLUSION

In this paper, we introduce DOMINO, a dual-system for multi-step multimodal reasoning. DOMINO alternates between two key modules, System-1 for targeted information extraction from images and System-2 for task decomposition and answer generation. We compare our model's performance against both supervised and pipeline approaches on different chart/plot question answering datasets, and achieve better or comparable results. Further analysis shows that: (1) A general description of the chart helps System-2 better at task decomposition. (2) DOMINO is more robust in terms of handling complex charts. (3) Training System-2 for better performance is data-efficient, but System-2 benefits differently from the skills acquired during fine-tuning.

## ETHICS STATEMENT

Step-by-step reasoning to derive an answer from large models builds transparency and trust for users, and eases bug-fixing. In this context, we hope our work builds transparency by providing the intermediate steps used to derive at an answer. However, similar to other works on question answering from charts, our models could possibly be abused to mislead the public about the charts content and implications. Although our models obtain comparable or state-of-the-art results on the datasets we evaluated, we can not guarantee that the output of these models will always be correct. We have shared our hyper-parameter settings in the paper to ensure the reproducibility of our experimental results and we will open source our code to Github.

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

# A  Appendix

## A.1  Training and Evaluation Datasets

### A.1.1  Dataset Statistics

We used the following datasets in this paper:

- ChartQA (Masry et al., 2022): A dataset of both human-authored and machine-generated questions about bar, line, and pie charts sourced from Statista (`statista.com`), The Pew research (`pewresearch.org`), Our World In Data or OWID (`ourworldindata.org`), and Organisation for Economic Co-operation and Development or OECD (`oecd.org`). The training set consists of $7,398$ human-authored questions over $3,699$ charts and $20,901$ machine-generated questions over $15,474$ charts.

- PlotQA (Methani et al., 2020): A dataset sourced from World Bank Open Data, Open Government Data, Global Terrorism Database which contain statistics about various indicator variables. The data contains positive integers, floating point values, percentages, and values on a linear scale, which range from 0 to 3.50e+15. The dataset consists of questions categorized into structural understanding, data retrieval and reasoning over bar plots, line plots, and scatter plots. This dataset does not consider any visual features of a chart (Masry et al., 2022).

- DVQA (Kafle et al., 2018): A synthetic dataset containing open ended questions about bar charts, where many questions refer to texts specific to corresponding charts.

- FigureQA (Kahou et al., 2018): A synthetic dataset containing yes/no questions about line graphs, bar and pie charts.

Dataset statistics of the test sets used in this paper are reported in the following table. For the synthetic datasets (i.e., PlotQA, DVQA, FigureQA), we randomly sample 10K examples for evaluation.

| Dataset | # Charts | # QA Pairs |
|---|---|---|
| ChartQA (Aug.) | 987 | 1250 |
| ChartQA (Human) | 625 | 1250 |
| PlotQA V1 | 8643 | 10000 |
| PlotQA V2 | 8252 | 10000 |
| DVQA (reasoning) | 9138 | 10000 |
| FigureQA Val1 | 5000 | 5000 |
| FigureQA Val2 | 5000 | 5000 |

### A.1.2  Templates for Data Generation for System-1

We use templates with ChartQA and PlotQA training sets to generate the data for fine-tuning System-1. Tables 5 and 6 show the templates we used and examples of the generated data, respectively.

Table 5: Templates for data generation for System-1.

| Atomic Operation | Template | Used Data |
|---|---|---|
| Describe | Let's describe the figure. | Use general figure info, color, text label and name from original annotations. |
| Extract-Point | Let's extract the data of <entity> BY <entity2>.
Let's extract the data of <entity>. | Retrieve the value of a cell in a table specified by the two entities as row or column. |
| Extract-Group | Let's extract the data of <entity>.
Let's extract all the values. | Use table header with values for the specified entity. If only one entity is present, then extract all values. |

Table 6: Examples of the atomic operations generated using templates for ChartQA and PlotQA.

| Atomic Operation | Source | Question | Answer |
|---|---|---|---|
| Describe | ChartQA | Let's describe the figure. | The figure shows the data of: Income in million U.S. dollars (blue). The x-axis shows: Taylor Swift \| Kylie Jenner \| Kanye West \| Lionel Messi \| Ed Sheeran \| Cristiano Ronaldo \| Neymar \| The Eagles \| Dr. Phil McGraw \| Canelo Alvarez. |
| Describe | PlotQA | Let's describe the figure. | The figure shows the data of: Fragile and conflict affected situations (grey) \| Iraq (brown) \| Moldova (orange). The x-axis shows: 2004 \| 2005 \| 2006 \| 2007 \| 2004 \| 2005 \| 2006 \| 2007. |
| Extract-Point | ChartQA | Let's extract the data of 2015. | The data is 296.0. |
| Extract-Point | PlotQA | Let's extract the data of Canada BY 1965. | The data is 20.82. |
| Extract-Group | ChartQA | Let's extract the data of Total market. | The data is 18 in 2019, 20.0 in 2018, 22.0 in 2017, 23.0 in 2016, 24.0 in 2015, 25.0 in 2014, 26.0 in 2013, 27.0 in 2012, 26.0 in 2011. |
| Extract-Group | PlotQA | Let's extract the data of 1994. | The data is 0.16 in Merchandise exports, 0.36 in Merchandise imports. |

A.1.3    EXAMPLE OF ANNOTATED DATA FOR SYSTEM-2

Examples of the collected annotated data for System-2 are presented bellow. These examples are from PlotQA. Sentences that are input to the LLM are wrapped with [INST] tags.

[INST] Q: In how many years, is the value of the bar greater than 851? [/INST]
A: Let's describe the figure.
[INST] The figure shows the data of: Neonatal deaths (green). The x-axis shows: 2000 | 2001 | 2002 | 2003 | 2004 | 2005. [/INST]
Let's extract the data of Neonatal deaths.
[INST] The data is 853 in 2000, 847 in 2001, 822 in 2002, 828 in 2003, 818 in 2004, 843 in 2005. [/INST]
The values that are greater than 851 are [853]. So the answer is 1.

---

[INST] Q: Across all years, what is the minimum pupil-teacher ratio in Costa Rica? [/INST]
A: Let's describe the figure.
[INST] The figure shows the data of: Least developed countries (blue) | Cameroon (purple) | Costa Rica (yellow) | Tajikistan (brown). The x-axis shows: 2000 | 2001 | 2006 | 2007 | 2008 | 2011. [/INST]
Let's extract the data of Costa Rica.
[INST] The data is 18.84 in 2000, 19.57 in 2001, 17.79 in 2006, 17.91 in 2007, 15.64 in 2008, 14.92 in 2011. [/INST]
The minimum value is 14.92 in 2011. So the answer is 14.92.

## A.2 EVALUATION PROMPTS

Below is the 1-shot prompt used by DePlot (Liu et al., 2023a), which includes one table followed by five question and answer pairs:

Read the table below to answer the following questions.
Header: Year | Democrats | Republicans | Independents
Row 1: 2004 | 68.1% | 45.0% | 53.0%
Row 2: 2006 | 58.0% | 42.0% | 53.0%
Row 3: 2007 | 59.0% | 38.0% | 45.0%
Row 4: 2009 | 72.0% | 49.0% | 60.0%
Row 5: 2011 | 71.0% | 51.2% | 58.0%
Row 6: 2012 | 70.0% | 48.0% | 53.0%
Row 7: 2013 | 72.0% | 41.0% | 60.0%

Q: In which year republicans have the lowest favor rate?
A: Let's find the column of republicans. Then let's extract the favor rates: [45.0, 42.0, 38.0, 49.0, 51.2, 48.0, 41.0]. The smallest number is 38.0, that's Row 3. Row 3 is year 2007. The answer is 2007.

Q: What is the sum of Democrats' favor rates of 2004, 2012, and 2013?
A: Let's find the rows of years 2004, 2012, and 2013. We find Row 1, 6, 7. The favor dates of Democrats on that 3 rows are 68.1, 70.0, and 72.0. 68.1+70.0+72=210.1. The answer is 210.1.

Q: By how many points do Independents surpass Republicans in the year of 2011?
A: Let's find the row with year = 2011. We find Row 5. We extract Independents and Republicans' numbers. They are 58.0 and 51.2. 58.0-51.2=6.8. The answer is 6.8.

Q: Which group has the overall worst performance?
A: Let's sample a couple of years. In Row 1, year 2004, we find Republicans having the lowest favor rate 45.0 (since 45.0<68.1, 45.0<53.0). In year 2006, Row 2, we find Republicans having the lowest favor rate 42.0 (42.0<58.0, 42.0<53.0). The trend continues to other years. The answer is Republicans.

Q: Which party has the second highest favor rates in 2007?
A: Let's find the row of year 2007, that's Row 3. Let's extract the numbers on Row 3: [59.0, 38.0, 45.0]. 45.0 is the second highest. 45.0 is the number of Independents. The answer is Independents.

Read the table below to answer the following questions.

Below is the alternate 5-shot prompt that we used for evaluating DePlot, which includes five tables with one question and answer pair for each:

Read the table to answer the following question.

Header: Entity | 2008 | 2009 | 2010 | 2011 | 2012 | 2013 | 2014
Row 1: Oman | 183.88 | 233.80 | 210.69 | 195.26 | 196.32 | 154.21 | 153.22
Row 2: Samoa | 40.72 | 40.04 | 39.21 | 40.63 | 41.47 | 41.76 | 42.77
Q: In which year the private health expenditure per person in Oman is 210.69?
A: Let's find the row of Oman, that's Row 1. Let's find the number 210.69, that's for the year of 2010. The answer is 2010.

Header: Year | NET Excellent/ good | NET Only fair/ poor
Row 1: Feb 2014 | 54 | 39
Row 2: June 2015 | 55 | 0
Row 3: Jan 2018 | 45 | 46
Row 4: Sept. 2018 | 331 | 62
Q: By how many points does NET Excellent/good surpass NET Only fair/poor in the year of 2014?
A: Let's find the row of 2014, that's Row 1. We extract NET Excellent/good's and NET Only fair/poor's numbers. They are 54.00 and 39.00. 54.00-39.00=15.00. The answer is 15.00.

Header: Characteristic | Share of respondents
Row 1: Very positive | 4%
Row 2: Fairly positive | 41%
Row 3: Fairly negative | 50%
Row 4: Very negative | 11%
Q: How many perceptions have a value below 40% in America?
A: Let's find the values that are below 40: [4, 11]. The answer is 2.

Header: Characteristic | Consoles | PC | Mobile and ancillary | Other
Row 1: 2020 | 2784 | 2056 | 2559 | 687
Row 2: 2019 | 1920 | 1718 | 2203 | 648
Row 3: 2018 | 2538 | 2180 | 2175 | 607
Row 4: 2017 | 2389 | 2042 | 2081 | 505
Row 5: 2016 | 2453 | 2124 | 1674 | 357
Row 6: 2015 | 2391 | 1499 | 418 | 356
Q: In 2020, how much money did Activision Blizzard's console segment generate in annual revenues in Australia?
A: Let's find the row of 2020, that's Row 1. Let's find the number for Consoles, that's 2784. The answer is 2784.

Header: Characteristic | Share of respondents
Row 1: Decreased | 81%
Row 2: No impact | 16%
Row 3: Increased | 3%
Q: Is the sum of two smallest segments greater than the largest segment?
A: Let's extract the numbers of all the rows: [81, 16, 3]. The two smallest numbers are 16 and 3. 16+3=19, which is smaller than 81. The answer is no.

Below is the 5-shot prompt we used for evaluating DOMINO on ChartQA:

Answer the following questions step by step.

Q: In which year the private health expenditure per person in Oman is 210.69?
A: Let's describe the figure.

The figure shows the data of: Oman (brown) | Samoa (dark blue). The x-axis shows: 2008 | 2009 | 2010 | 2011 | 2012 | 2013 | 2014.
Let's extract the data of Oman.
The data is 183.88 in 2008, 233.80 in 2009, 210.69 in 2010, 195.26 in 2011, 196.32 in 2012, 154.21 in 2013, 153.22 in 2014.
The value 210.69 is in year 2010. So the answer is 2010.

Q: By how many points does NET Excellent/good surpass NET Only fair/poor in German in the year of 2018?
A: Let's describe the figure.
The figure shows the data of: NET Excellent/ good (blue) | NET Only fair/ poor (orange). The x-axis shows: Brazil, German, Russia, U.S., Japan.
Let's extract the data of NET Excellent/ good BY German.
The data is 54.00.
Let's extract the data of NET Only fair/ poor BY German.
The data is 39.00.
54.00 surpasses 39.00 by 54.00-39.00=15.00. So the answer is 15.00.

Q: How many perceptions have a value below 40% in America?
A: Let's describe the figure.
The figure shows the data of: Share of respondents (blue). The x-axis shows: Very positive, Fairly positive, Fairly negative, Very negative.
Let's extract the data of Share of respondents.
The data is 4.00 in Very positive, 41.00 in Fairly positive, 50.00 in Fairly negative, 11.00 in Very negative. The values that are below 40.00 are [4.00, 11.00]. So the answer is 2.

Q: In 2020, how much money did Activision Blizzard's console segment generate in annual revenues in Australia? A: Let's describe the figure.
The figure shows the data of: Consoles (blue) | PC* (dark blue) | Mobile and ancillary** (grey) | Other (dard red). The x-axis shows: 2019 | 2020 | 2021 | 2022.
Let's extract the data of Consoles BY 2020.
The data is 2784.00.
The Console segment generated 2784.00 in annual revenues in 2020. So the answer is 2784.00.

Q: Is the sum of two smallest segments greater than the largest segment?
A: Let's describe the figure.
The figure shows the data of: Value. The x-axis shows: Decreased | No impact | Increased.
Let's extract the data of Value.
The data is 81.00 in Decreased, 16.00 in No impact, 3.00 in Increased.
Among [81.00, 16.00, 3.00], the two smallest values are 16.00 and 3.00 while the largest value is 81.00. 16.00+3.00=19.00, which is smaller than 81.00. So the answer is no.

Below is the 5-shot prompt we used for evaluating DOMINO on ChartQA without `Describe`:

Answer the following questions step by step.

Q: In which year the private health expenditure per person in Oman is 210.69?
A: Let's extract the data of Oman.
The data is 183.88 in 2008, 233.80 in 2009, 210.69 in 2010, 195.26 in 2011, 196.32 in 2012, 154.21 in 2013, 153.22 in 2014.
The value 210.69 is in year 2010. So the answer is 2010.

Q: By how many points does NET Excellent/good surpass NET Only fair/poor in German in the year of 2018?

A: Let's extract the data of NET Excellent/ good BY German.
The data is 54.00.
Let's extract the data of NET Only fair/ poor BY German.
The data is 39.00.
54.00 surpasses 39.00 by 54.00-39.00=15.00. So the answer is 15.00.

Q: How many perceptions have a value below 40% in America?
A: Let's extract the data of Share of respondents.
The data is 4.00 in Very positive, 41.00 in Fairly positive, 50.00 in Fairly negative, 11.00 in Very negative. The values that are below 40.00 are [4.00, 11.00]. So the answer is 2.

Q: In 2020, how much money did Activision Blizzard's console segment generate in annual revenues in Australia? A: Let's extract the data of Consoles BY 2020.
The data is 2784.00.
The Console segment generated 2784.00 in annual revenues in 2020. So the answer is 2784.00.

Q: Is the sum of two smallest segments greater than the largest segment?
A: Let's extract the data of Value.
The data is 81.00 in Decreased, 16.00 in No impact, 3.00 in Increased.
Among [81.00, 16.00, 3.00], the two smallest values are 16.00 and 3.00 while the largest value is 81.00. 16.00+3.00=19.00, which is smaller than 81.00. So the answer is no.

## A.3   ERROR EXAMPLES

Table 7: Number of errors per template type for PlotQA V2 (examples follow). Numbers in parenthesis indicate total number of examples per template type in the 10K sample we evaluated.

| Method | data retrieval (1379) | structural (447) | arithmetic (5147) | compound (637) | comparison (1815) | min-max (575) |
|---|---|---|---|---|---|---|
| **Few-Shot DePlot** | | | | | | |
| LLaMA-2 (70B) | 547 | 275 | 3448 | 323 | 757 | 185 |
| **DOMINO (our method)** | | | | | | |
| LLaMA-2 (70B) | 400 | 203 | 2928 | 193 | 312 | 64 |
| Fine-tuned LLaMA-2 (70B) | 388 | 189 | 1861 | 177 | 242 | 62 |
| Fine-tuned LLaMA-2 (70B) (SC) | 377 | 159 | 915 | 184 | 233 | 63 |

Example of errors for each template type in PlotQA v2 made by DOMINO with finetuned LLaMA-2 (70B) is illustrated in the following table.

| Template | Question | Answer | Prediction |
|---|---|---|---|
| Continued on next page | | | |

| Template | Question | Answer | Prediction |
|---|---|---|---|
| Data retrieval | What is the label of the 2nd group of bars from the top ? 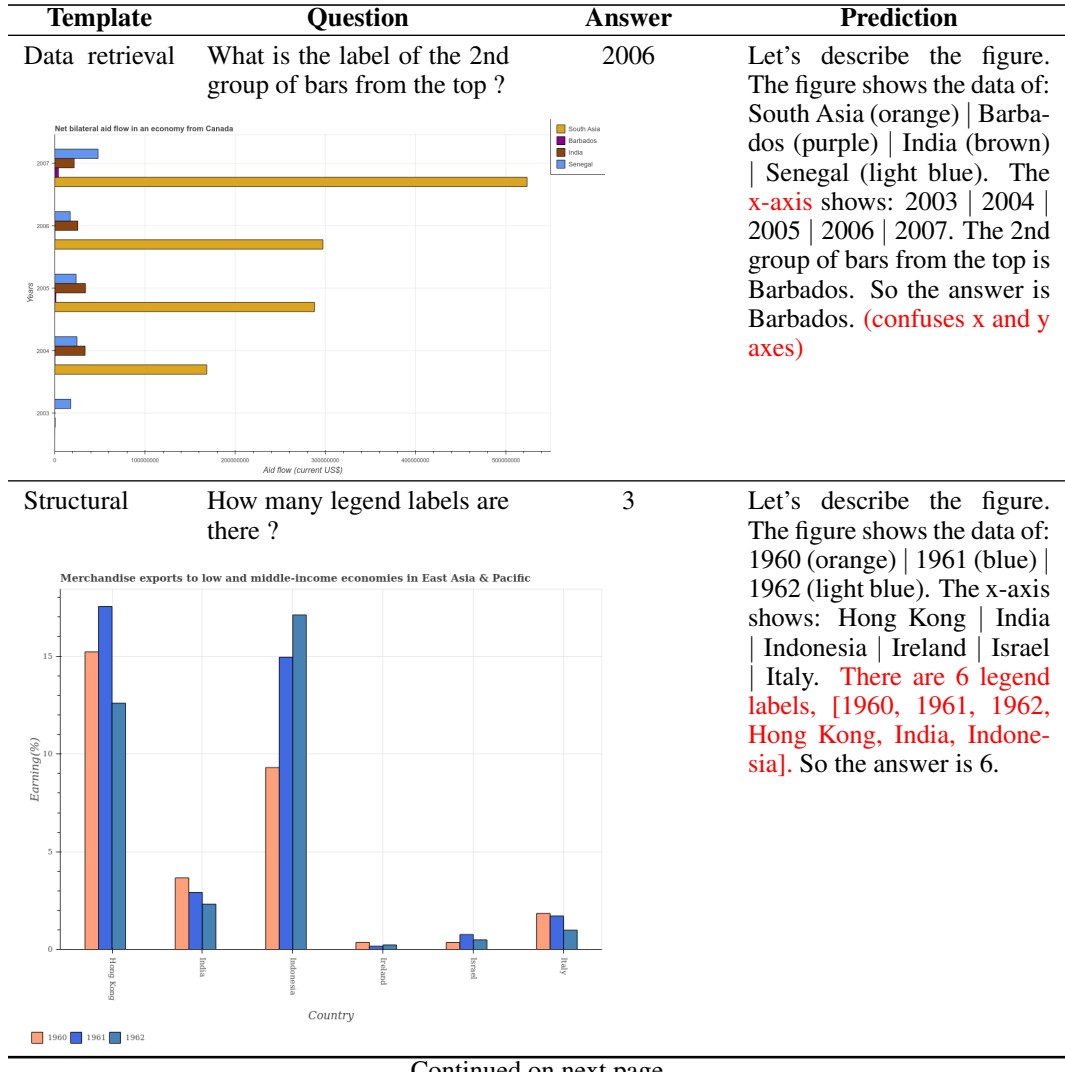 | 2006 | Let's describe the figure. The figure shows the data of: South Asia (orange) \| Barbados (purple) \| India (brown) \| Senegal (light blue). The x-axis shows: 2003 \| 2004 \| 2005 \| 2006 \| 2007. The 2nd group of bars from the top is Barbados. So the answer is Barbados. (confuses x and y axes) |
| Structural | How many legend labels are there ? | 3 | Let's describe the figure. The figure shows the data of: 1960 (orange) \| 1961 (blue) \| 1962 (light blue). The x-axis shows: Hong Kong \| India \| Indonesia \| Ireland \| Israel \| Italy. There are 6 legend labels, [1960, 1961, 1962, Hong Kong, India, Indonesia]. So the answer is 6. |

| Template | Question | Answer | Prediction |
|---|---|---|---|
| Arithmetic | What is the difference between the Payments (BoP, current US$) of Aruba in 2008 and the Payments (BoP, current US$) of Philippines in 2009? | -50752953286.0 | Let's describe the figure. The figure shows the data of: Aruba (brown) │ Indonesia (purple) │ Philippines (brown)│ Serbia (pink). The x-axis shows: 2007 │ 2008 │ 2009 │ 2010 │ 2011 │ 2012. Let's extract the data of Aruba BY 2008. The data is 7056000000.0. Let's extract the data of Philippines BY 2009. The data is 57792000000.0. The difference is 7056000000.0-57792000000.0=-47232000000.0. So the answer is -47232000000.0. |
| Compound | In how many years, is the percentage of male labor force with secondary education in Lithuania greater than the average percentage of male labor force with secondary education in Lithuania taken over all years ? | 4 | Let's describe the figure. The figure shows the data of: Bulgaria (green) │ Hong Kong (brown) │ Lithuania (green) │ Paraguay (green). The x-axis shows: 2008 │ 2009 │ 2010 │ 2011 │ 2012 │ 2013.\nLet's extract the data of Lithuania. The data is 64.0 in 2008, 64.5 in 2009, 64.8 in 2010, 64.4 in 2011, 64.7 in 2012, 63.3 in 2013. The average percentage of male labor force with secondary education in Lithuania is (64.0+64.5+64.8+64.4+64.7+63.3)/6=64.2. 64.0 is not greater than 64.2, 64.5 is not greater than 64.2, 64.8 is greater than 64.2, 64.4 is not greater than 64.2, 64.7 is not greater than 64.2, 63.3 is not greater than 64.2. So the answer is 1. |

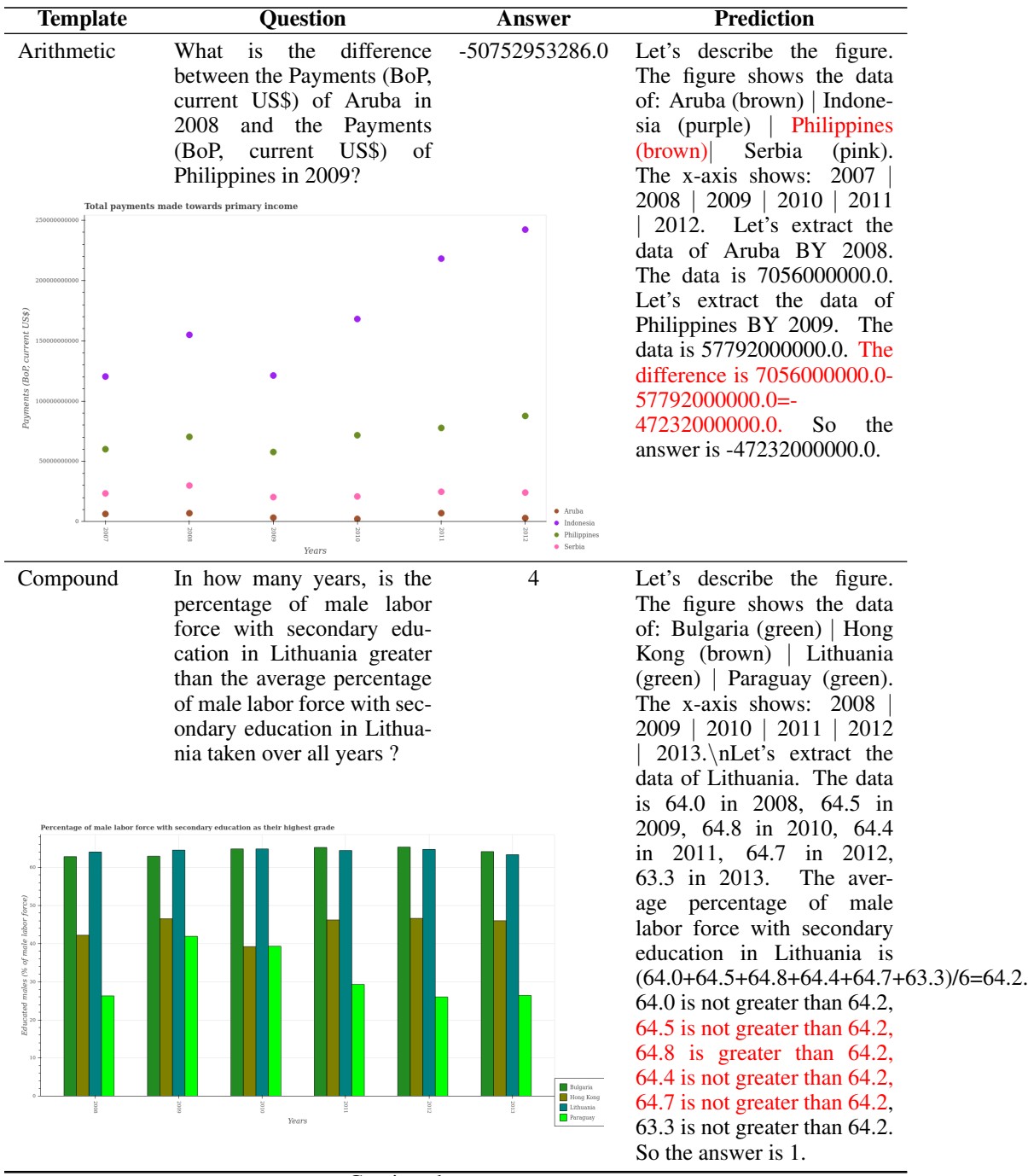

| Template | Question | Answer | Prediction |
|---|---|---|---|
| Comparison | What is the ratio of the Aid flow (current US$) of Portugal in 1972 to that in 1973 ?  | 0.5217 | Let's describe the figure. The figure shows the data of: Norway (pink) \| Portugal (orange) \| Sweden (green) \| Switzerland (brown). The x-axis shows: 1972 \| 1973 \| 1974 \| 1975 \| 1976 \| 1977 \| 1978 \| 1979.\nLet's extract the data of Portugal BY 1972. The data is 1000000.0.\nLet's extract the data of Portugal BY 1973. The data is 1290000.0. The ratio is 1000000.0/1290000.0=0.7752. So the answer is 0.7752. (requires checking the numbers against the y-axis) |
| Min-Max | Across all countries, what is the minimum total arable land in 1963 ?  | 110000.0 | Let's describe the figure. The figure shows the data of: 1961 (orange) \| 1962 (blue) \| 1963 (dark green). The x-axis shows: Gabon \| Gambia, The \| Germany \| Ghana \| Greece. Let's extract the data of 1963. The data is 100000.0 in Gabon, 120000.0 in Gambia, The, 12170000.0 in Germany, 1700000.0 in Ghana, 3050000.0 in Greece. The minimum value is 100000.0 in Gabon. So the answer is 100000.0. (requires checking the numbers against the y-axis) |

Example of errors about task decomposition in ChartQA-human made by DOMINO with finetuned LLaMA-2 (70B) is illustrated in the following table.

| Question | Answer | Prediction |
|---|---|---|
| Continued on next page | | |

| Question | Answer | Prediction |
|---|---|---|
| How many people use daily? 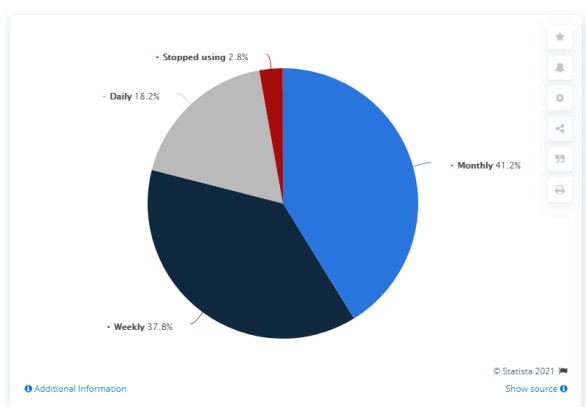 | 18.2 | Let's describe the figure. The figure shows the data of: Value (blue). The x-axis shows: Weekly \| Daily \| Stopped using. Let's extract the data of Value. The data is 41.2 in Monthly, 37.8 in Weekly, 18.2 in Daily, 2.8 in Stopped using. Among [41.2, 37.8, 18.2, 2.8], the value that is 18.2 is the number of people who use daily. So the answer is 18.2. (fails to request the only information required for answering although the model still answers correctly) |
| What's the percentage of people who don't believe it can impact us and don't know much about covid? 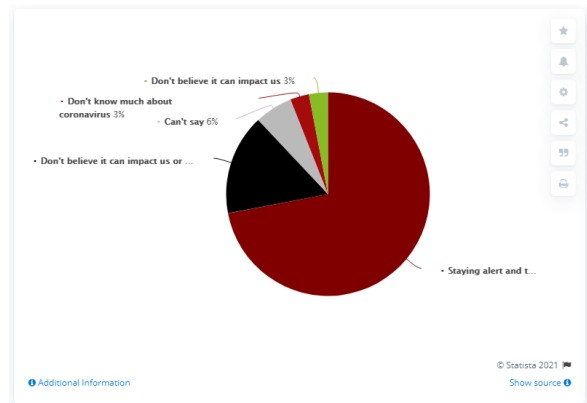 | 6 | Let's describe the figure. The figure shows the data of: Value (dark blue). The x-axis shows: Staying alert and to the ground \| Don't believe it can impact us or \| Don't know much about. Let's extract the data of Value. The data is 1.0 in Staying alert and the 2019. The percentage of people who don't believe it can impact us and don't know much about covid is 1.0. So the answer is 1.0. (invalid queries for the vision module) |

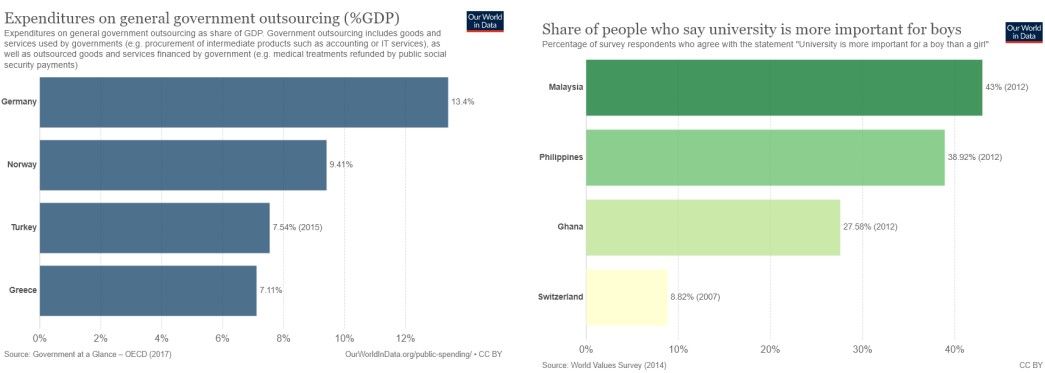

Figure 4: The charts for the case study in § 6. The charts are from ChartQA (Masry et al., 2022).

