# OpenReview forum: "DOMINO: A Dual-System for Multi-step Visual Language Reasoning"
_ICLR.cc/2024/Conference — Submitted to ICLR 2024_

### Official Review · Reviewer_j7sH · 2023-10-31

**Soundness:** 2 fair
**Presentation:** 2 fair
**Contribution:** 2 fair
**Rating:** 5
**Confidence:** 4

**Summary:**

The paper focus on visual language reasoning problems which requires extraction of text or numbers from information-dense images like charts or plots. The proposed method includes a dual-system for multi-step multimodal reasoning, which consists of a “System-1” step for visual information extraction and a “System-2” step for deliberate reasoning. By fine-tuning LLaMA-2 70B on only a small amount of data on multi-step reasoning, the accuracy of the model surpasses the best fully-supervised end-to-end approach by 5.7% and a pipeline approach with FlanPaLM (540B) by 7.5% on ChartQA.

**Strengths:**

• The paper is well written and easy to understand. Figure 1 provides a good overview of the complete system.
• The paper presents promising results on ChartQA and outperforms prior supervised baselines.
• The paper includes ablation studies in Figure 3.

**Weaknesses:**

• Novelty: The core idea of the paper is very similar to prior work, including “Visual Programming: Compositional Visual Reasoning Without Training, CVPR 2023” which also uses a large LLM for reasoning and perception modules to extract information from images. Additionally, “Socratic Models: Composing Zero-Shot Multimodal Reasoning with Language, arXiv 2022” also performs zero-shot multi-modal reasoning in a similar fashion. “Look, Remember and Reason: Visual Reasoning with Grounded Rationales, ICML workshop 2023” combines System-1 and System-2 inference in a single model using rationales.

• It is unclear why the performance on PlotQA much worse compared to ChartQA. The paper mentions that PlotQA “is a synthetic dataset with template based and restricted types of questions”. But this should be easier to solve compared to ChartQA, as the proposed approach also follows templated reasoning steps. The paper should make it clear with ample qualitative examples why performance on PlotQA is lacking.

• Fairness of the comparison to Few-Shot DePlot versions of GPT-4 and LLaMA: The proposed  DOMINO version of LLaMA has more information about the chart in question. Therefore, it is unclear if the evaluation is fair.

• Qualitive examples: The paper is lacking qualitive examples from PlotQA in the main paper. The main paper only includes a single qualitative example from ChartQA in the main paper. Examples of failure cases in Table 7 are hard to follow as the associated charts are not available. The paper should include more quantitative examples  which are easier to follow. The format of “GPT-4 Technical Report, arXiv 2023” can serve as a guiding example.

• Additional datasets: The paper evaluates performance only on two datasets. There are also more challenging datasets available: SciCap (http://scicap.ai/). The SciCap uses real-world data and requires high-level reasoning along with low-level understanding of scientific figures. It would be an ideal testbed to evaluate the performance of the proposed approach.

**Questions:**

• The paper should discuss prior work such as “Visual Programming: Compositional Visual Reasoning Without Training, CVPR 2023”, “Socratic Models: Composing Zero-Shot Multimodal Reasoning with Language, arXiv 2022”, “Look, Remember and Reason: Visual Reasoning with Grounded Rationales, ICML workshop 2023” in more detail.

• The should discuss the challenges associated with PlotQA in more detail, ideally with qualitative examples.

• The fairness of the comparison to Few-Shot DePlot versions of GPT-4 and LLaMA should be discussed in more detail.

---

> ### Author Response · Authors · 2023-11-17
> **Response to Reviewer j7sH**
>
> Thank you for the comments and suggestions. Answers to your questions as well as clarifications to some of your comments below:
>
> > Novelty: The core idea of the paper is very similar to prior work, including “Visual Programming: Compositional Visual Reasoning Without Training, CVPR 2023”, “Socratic Models: Composing Zero-Shot Multimodal Reasoning with Language, arXiv 2022” and “Look, Remember and Reason: Visual Reasoning with Grounded Rationales, ICML workshop 2023”.
>
> DOMINO differs from these works in how the interactions between the language and vision modules are decided. Both “Socratic Models” and “Look, Remember and Reason” pre-define the interactions with templates, which do not generalize to questions with new reasoning processes. “Visual Programming” decides the whole interaction process by generating a program, but the LM does not get to react differently based on the intermediate results returned from the vision module, which limits the question types that could be solved. By contrast, DOMINO learns to compose the atomic operations on the fly based on both the question and the intermediate results from the vision module, which is more flexible.
>
>
> > It is unclear why the performance on PlotQA much worse compared to ChartQA. The paper mentions that PlotQA “is a synthetic dataset with template based and restricted types of questions”. But this should be easier to solve compared to ChartQA, as the proposed approach also follows templated reasoning steps. The paper should make it clear with ample qualitative examples why performance on PlotQA is lacking.
>
> The charts in PlotQA involve extremely large numbers ranging from 0 to 3:50e+15, which pose a challenge to reasoning for both the vision and language models. The fully-supervised method leverages the sufficiently large training set (with over 20M examples) to learn the data bias, which is also pointed out in the DePlot paper (Liu et al., 2023).
>
> > Fairness of the comparison to Few-Shot DePlot versions of GPT-4 and LLaMA: The proposed DOMINO version of LLaMA has more information about the chart in question. Therefore, it is unclear if the evaluation is fair.
>
> The few-shot DePlot method includes the entire table associated with the chart, whereas the DOMINO version of LLaMA does not include this information (as illustrated in Table 4) and as such does not necessarily include more information about the chart. We agree that a table is not an authentic representation of the chart, and this is one of the motivations for our work. DOMINO may include other types of information (e.g., color) and can also ask for additional information by interacting with the vision module.
>
> > Qualitative examples: The paper is lacking qualitative examples from PlotQA in the main paper. The main paper only includes a single qualitative example from ChartQA in the main paper. Examples of failure cases in Table 7 are hard to follow as the associated charts are not available. The paper should include more quantitative examples which are easier to follow. The format of “GPT-4 Technical Report, arXiv 2023” can serve as a guiding example.
>
> Due to the page limit, we only include two examples in Table 4 in the main paper and attach the associated charts in Figure 4 in the Appendix. We have included the associated charts for the examples in Table 7 in the updated draft for your reference and we will move one PlotQA example to the main paper.
>
> > Additional datasets: The paper evaluates performance only on two datasets. There are also more challenging datasets available: SciCap (http://scicap.ai/). The SciCap uses real-world data and requires high-level reasoning along with low-level understanding of scientific figures. It would be an ideal testbed to evaluate the performance of the proposed approach.
>
> Besides ChartQA and PlotQA, we also conducted experiments on two out-of-distribution datasets including DVQA and FigureQA as shown in Table 2 in the main paper. We focused on visual language reasoning in this work since it requires the LM to really rely on the visual information to answer a question about a chart and conduct multi-step reasoning while in other VQA tasks, the LM either can rely on language prior (but not necessarily the image) or does not need to conduct multi-step reasoning to answer a question. Thus, SciCap, which is actually a caption generation task, might not be an ideal testbed for our problem. Based on the examples shown in SciCap, some information mentioned in the target caption is not even grounded by the images. But we view SciCap as a potentially good resource for training our vision module to conduct the Describe operation. Thank you for your suggestion.

---

> > ### Comment · Reviewer_j7sH · 2023-11-20
> > **Response to Rebuttal**
> >
> > Thanks for the detailed response. However, my major concerns remain:
> >
> > 1) Novelty: The paper is nice application of prior works especially: “Socratic Models: Composing Zero-Shot Multimodal Reasoning with Language, arXiv 2022”. However, the paper provides limited new insights.
> >
> > 2) Efficiency: Related to the point above, I do not believe that the presented system will gain much traction in the future, due to the multiple calls to different LLMs leading to poor efficiency (also pointed out by Reviewer CNjW). Current SOTA LLM such as GPT-4V already shows promising performance on similar tasks without the need of multiple function calls (https://arxiv.org/pdf/2309.17421.pdf).
> >
> > 3) Results on PlotQA: The current results show that the method is brittle even after fine-tuning.
> >
> > I will keep my score.

---

> > > ### Author Response · Authors · 2023-11-22
> > >
> > > Thank you for your quick response! Please find below our clarifications which should address your remaining concerns.
> > >
> > > 1. Novelty
> > >
> > > The distinction between our method and "Socratic Models" is that we do not need to **pre-define the interactions with templates** while  "Socratic Models" does. Thus, our method is not an application of " "Socratic Models".
> > >
> > > 2. Efficiency
> > >
> > > We want to clarify that although there may be multiple calls to the vision module in DOMINO, we do not need to do the forward passing in LLM from scratch whenever feeding the intermediate results since we cache the previous hidden states (keys and values). The benefit of offloading the step of visual information extraction to a vision module is to **reduce the hallucination behavior of LLM**, which is a well-known issue. Moreover, DOMINO is actually more efficient compared to the few-shot DePlot model due to two reasons: (1) The vision module in DOMINO only needs to generate the required information based on the image while the vision module in DePlot needs to generate the whole table, which can be arbitrarily long. (2) As a result of (1), DOMINO does not need to take as input the whole table sequence which would consume a large part of the context window in the LM of DePlot.
> > >
> > > 3. Results on PlotQA
> > >
> > > We want to highlight that our method is more **data-efficient** compared with fine-tuning Deplot as illustrated in Figure 3. With different numbers (20-100) of training examples on multi-step reasoning, fine-tuned DOMINO generally outperforms fine-tuned Deplot. As for the best method MATCHA, it achieves 90.7% accuracy on PlotQA-V2 by learning from over 20M examples while our method achieves the second-best result (80.7%) with only 100 training examples. We would appreciate it if you could

---

### Official Review · Reviewer_CNjW · 2023-10-31

**Soundness:** 2 fair
**Presentation:** 3 good
**Contribution:** 3 good
**Rating:** 5
**Confidence:** 3

**Summary:**

This paper introduces DOMINO, a dual-system designed for charts/plots reasoning. DOMINO consists of two models: The first model, called system-1, uses vision and language to extract specific information from images. The second model, system-2, is a large language model that decomposes tasks and generates answers. Experimental results indicate that DOMINO surpasses traditional pipeline approaches in handling both in- and out-of-distribution data. With limited training samples, DOMINO also achieves SOTA results on ChartQA.

**Strengths:**

1. This method is intuitive, and I am happy to see the introduction of dual-system into vision-language reasoning.
2. The proposed method achieves SOTA results on ChartQA.
3. Analysis shows that DOMINO is more robust in handling complex charts.

**Weaknesses:**

1. The author didn't discuss about the efficiency. How does the inference efficiency of DOMINO compare to the baseline method?
2. The template seems relatively limited, more non-chartQA tasks are needed to confirm the potential of this method.

**Questions:**

1. What types of charts are included in ChartQA and PlotQA? I think adding relevant descriptions can help people have a more intuitive understanding of the capabilities of this method.
2. Does the author consider the dual-system approach to be universally applicable? Can it replace other MLLM methods (such as BLIP2 [1], LLAVA [2]) and become a common solution for solving visual QA problems? For example, besides tasks like chartQA, can DOMINO also generalize to other tasks (such as VQA)?

[1] Li, J., Li, D., Savarese, S., & Hoi, S.C. (2023). BLIP-2: Bootstrapping Language-Image Pre-training with Frozen Image Encoders and Large Language Models. ArXiv, abs/2301.12597.
[2] Liu, H., Li, C., Wu, Q., & Lee, Y.J. (2023). Visual Instruction Tuning. ArXiv, abs/2304.08485.

---

> ### Author Response · Authors · 2023-11-17
> **Response to Reviewer CNjW**
>
> Thank you for your comments and feedback. Please find below the answers to your questions:
>
> **W1: The author didn't discuss about the efficiency. How does the inference efficiency of DOMINO compare to the baseline method?**
>
> Although there may be multiple calls to the vision module in DOMINO,  DOMINO is more efficient compared to the few-shot DePlot model due to two reasons: (1) The vision module in DOMINO only needs to generate the required information based on the image while the vision module in DePlot needs to generate the whole table, which can be arbitrarily long. (2) As a result of (1), DOMINO does not need to take as input the whole table sequence which would consume a large part of the context window in the LM of DePlot.
> The supervised end-to-end method is more efficient in inference than both DePlot and DOMINO. However, the method generates the answers directly which is prone to error and lacks interpretability for complex questions that require multi-step reasoning. We have included this discussion in Section 6 of the paper.
>
> **W2: The template seems relatively limited, more non-chartQA tasks are needed to confirm the potential of this method.**
>
> We agree that DOMINO can be applied to other VQA tasks. In this work, we focused on visual language reasoning as it requires accurate information extraction from an image in addition to strong numerical/logical reasoning skills to answer a question based on the extracted information, as also discussed in previous works such as (Liu et al., 2023). As such, we compiled the minimum number of templates that could generally be used across different chart/plot types.
>
> **Q1: What types of charts are included in ChartQA and PlotQA? I think adding relevant descriptions can help people have a more intuitive understanding of the capabilities of this method.**
>
> The chart types in ChartQA include bar, line and pie charts. PlotQA has bar, line and dot-line charts. Thanks for your suggestion. We have expanded the descriptions of the experimented datasets in Appendix A.1.1 to include this information.
>
> **Q2: Does the author consider the dual-system approach to be universally applicable? Can it replace other MLLM methods (such as BLIP2 [1], LLAVA [2]) and become a common solution for solving visual QA problems? For example, besides tasks like chartQA, can DOMINO also generalize to other tasks (such as VQA)?**
>
> Yes, DOMINO can be applied to other VQA tasks since the LM can always use language as the vehicle for reasoning and guide the vision module to obtain the required information. We focused on visual language reasoning in this work since it requires the LM to really rely on the visual information to answer a question about a chart and conduct multi-step reasoning while in other VQA tasks, the LM either can rely on language prior (but not necessarily the image) or does not need to conduct multi-step reasoning to answer a question.

---

> > ### Comment · Area_Chair_bqLx · 2023-11-22
> >
> > Hi Reviewer CNjW,
> >
> > Pls read and reply to the authors' response.
> >
> > Thanks,
> > AC

---

> > ### Comment · Reviewer_CNjW · 2023-11-22
> > **Response to Rebuttal**
> >
> > I do appreciate the authors' time in responding to my comments. But after reading the response, some of my concerns have not been resolved:
> >
> > * Efficiency: Although the end-to-end baselines are weaker in performance compared to DOMINO, they have smaller parameter sizes and better inference efficiency than DOMINO. So when the parameter size of the end-to-end baselines is expanded to the same scale as DOMIMNO, can DOMINO still outperform these baselines? I hope the author can further improve the inference efficiency of DOMINO and add detailed efficiency comparisons.
> >
> > * Generalizability: I agree the dual-system approach can be generalized to other tasks, but the experiments in this paper have not yet supported this. I hope the author can expand this method to more multimodal tasks in the future.
> >
> > Thus, I will keep my original score.

---

> > > ### Author Response · Authors · 2023-11-22
> > >
> > > Thank you for your response! We agree that extending our idea to other multi-model tasks would be interesting for future work. As for efficiency, we want to clarify that although there may be multiple calls to the vision module in DOMINO, we **do not need to do the forward passing in LLM from scratch** whenever feeding the intermediate results since we cache the previous hidden states (keys and values). By offloading the step of visual information extraction to a vision module, we **prevent the LLM from hallucinating the visual information** (which is a well-known issue) and thus guarantee that the reasoning process is grounded (while the end-to-end models do not even provide the reasoning process). We hope you can take this advantage into account when evaluating our method. Thanks!

---

### Official Review · Reviewer_AQ48 · 2023-10-31

**Soundness:** 4 excellent
**Presentation:** 4 excellent
**Contribution:** 3 good
**Rating:** 8
**Confidence:** 3

**Summary:**

The paper proposes a two component system for chart/plot reasoning. The system is composed of a DePlot backbone and a LLaMa-2 model, the former used to extract information from the chart, while the later for decomposing the question and give final answer based on reasoning. After fine-tuning DePlot on instruction level tasks and LLM on a small number of hand written solutions, the system surpasses prompt-based baselines and some supervised methods. The performance gain is attributed to the improvement in both decomposition and answering.

**Strengths:**

1. The paper is clearly written.
2. The results are great, compared to few-shot baselines, and the performance gain is analyzed carefully.
3. The paper proposed a demonstration of two stage reasoning using LLMs for task decomposition using the feedback from perception results, which is novel compared to similar LLM-guided systems without feedback, e.g., [1]. The efficiency of fine-tuning of LLM also supports the decomposition of System-1/2.
4. The authors thoroughly discussed the functionality of each component in the reasoning process through ablation studies and analyzed the error made by the models.

[1] https://arxiv.org/abs/2211.11559

**Weaknesses:**

A few unclear points are raised in Questions.

**Questions:**

1. Why some of the results are not shown in Table 1?
2. Why is the correct answer for arithmetic in Table 7 is -50752953286.0?
3. What is the evaluation prompt used when there is no `Describe` step?
4. It would be great if the authors could discuss the applicability of the proposed method on other VQA tasks, e.g. CLEVR.
5. Is there any examples of the model failed at decomposing the problem?

---

> ### Author Response · Authors · 2023-11-17
> **Response to Reviewer AQ48**
>
> Thank you for your comments and feedback. Please find below the answers to your questions:
>
> **Q1. Why some of the results are not shown in Table 1?**
>
> The results for the baseline models are from their respective papers which are cited in the table. These works have not provided the results for all the datasets and/or dataset splits. We will clarify this in the paper. For GPT-4, we did not run the model on the much larger PlotQA evaluation sets due to high cost and usage limits as pointed out in the footnote on page 7.
>
> **Q2: Why is the correct answer for arithmetic in Table 7 is -50752953286.0?**
>
> We apologize for not including the original charts for each example in the paper and will include them in the updated version. The correct answer is from the PlotQA dataset. The question-answer pairs in PlotQA are generated by templates automatically and PlotQA considers the “difference” between A and B just as $A-B$ instead of the absolute difference. That is why the answer is negative in Table 7.
>
> **Q3: What is the evaluation prompt used when there is no Describe step?**
>
> In this case, the reasoning process starts directly with task decomposition and not the Describe step. For example, the first demonstration in the prompt would be:
> ```
> Q: In which year the private health expenditure per person in Oman is 210.69?
> A: Let’s extract the data of Oman.
> The data is 183.88 in 2008, 233.80 in 2009, 210.69 in 2010, 195.26 in 2011, 196.32 in 2012,
> 154.21 in 2013, 153.22 in 2014.
> The value 210.69 is in year 2010. So the answer is 2010.
> ```
> We have added the prompt used for this case to Appendix A-2.
>
> **Q4: It would be great if the authors could discuss the applicability of the proposed method on other VQA tasks, e.g. CLEVR.**
>
> Our proposed method can be applied to other VQA tasks given an appropriate vision module. We focused on visual language reasoning to necessitate the multiple interactions between language and vision modules since other VQA tasks either do not require multi-step reasoning or can be accomplished by language prior alone. We will highlight the flexibility of our method in the final draft.
>
> **Q5: Is there any examples of the model failed at decomposing the problem?**
>
> There are examples of incorrect decomposition due to ambiguity in the question. In some cases, the model fails to request the only information required for answering although the model still answers correctly finally:
> ```
> Q: How many people use daily?
> A: Let's describe the figure.
> The figure shows the data of: Value (blue). The x-axis shows: Weekly | Daily | Stopped using.
> Let's extract the data of Value.
> The data is 41.2 in Monthly, 37.8 in Weekly, 18.2 in Daily, 2.8 in Stopped using.
> Among [41.2, 37.8, 18.2, 2.8], the value that is 18.2 is the number of people who use daily. So the answer is 18.2.
> ```
> In other cases, the incorrect decomposition leads to invalid queries for the vision module. The vision module then gives the wrong intermediate result which finally leads to an incorrect answer.
> ```
> Q: What's the percentage of people who don't believe it can impact us and don't know much about covid?
> A: Let's describe the figure.
> The figure shows the data of: Value (dark blue). The x-axis shows: Staying alert and to the ground | Don't believe it can impact us or | Don't know much about.
> Let's extract the data of Value.
> The data is 1.0 in Staying alert and the 2019.
> The percentage of people who don't believe it can impact us and don't know much about covid is 1.0. So the answer is 1.0.
> ```
> We will extend our error analysis in the revised draft to include such examples.

---

> > ### Comment · Reviewer_AQ48 · 2023-11-21
> > **Response to rebuttal**
> >
> > Thanks for the detailed response. Though the other two reviewers raised concerns about efficiency, I do believe that retrieval is a critical component when the scale of the data is large and hope the authors could provide some examples. I will keep my score.

---

### Author Response · Authors · 2023-11-17
**Reviewer comments incorporated in newest revision**

We thank all reviewers for their helpful suggestions. Based on your suggestions, we have uploaded a new version of the paper that incorporates most of the reviewer comments. These include:
- Adding more description about ChartQA and PlotQA datasets to Appendix A.1.1;
- Adding charts to examples of failure cases in Table 7;
- Adding additional examples of model failure at decomposing the problem to examples of failure cases in Appendix  A.3;
- Adding evaluation prompt when there is no Describe step to Appendix A.2;
- Including a discussion of inference efficiency in Section 6 under Inference Efficiency;

---

### Meta-Review · Area_Chair_bqLx · 2023-12-05

**Metareview:**

In terms of strengths, reviewers found this paper to be easy-to-understand, achieved good results, and had careful analyses.

Overall, however, the majority of reviewers (including this AC) felt that this work was below the acceptace threshold for a selective/competitive venue like ICLR.

Key weaknesses are the limited novelty and generality, as well as unclear lasting contributions. In terms of novelty, reviewers pointed to other existing works that broadly followed the same approach. The authors' rebuttal relied on fairly subtle differences to make the case for novelty. Even if the authors' responses are fully accepted, they certainly did not make any case at all for strong novelty.

In this AC's view, even if properly appreciating this paper as a "system paper" that brings together off-the-shelf blackboxes, it is unclear how novel and general the proposed method is. "Dual system" approaches (and personally I view the use of system-1 and system-2 here as slight misuse/trivializing of these terms) are not new, and judging from the Appendix, the templates (e.g. "let's extract" or "let's describe) are specific to charts/graphs, meaning that the approach has limited generality. Overall, it is hard to imagine much lasting contribution to the broader ICLR community. A more focused or niche publication venue might be more appropriate.

(Minor point: I strongly urge the authors to revisit the title, which I find to be overly broad. Other similar published papers qualify the use of "visual language reasoning" by making it clear in the title that the work applies to chart/graphs/plots.)

In summary, this work is below the standards for acceptance to ICLR in its current form.

**Justification For Why Not Higher Score:**

Limited novelty and generality of the proposed method.

**Justification For Why Not Lower Score:**

N/A

---

### Decision · Program_Chairs · 2024-01-16

Reject